# Comparative Factors Influencing Entrepreneurial Skills Acquisition amongst Students in Rural Universities of Sub-Sahara Africa's Developing Nations

**Omotosho Ademola Olumuyiwa *** , **Kimanzi Matthew Kimweli** and **Motalenyane Alfred Modise**

Department of Languages and Social Sciences Education, Central University of Technology,
Bloemfontein 9301, Free State, South Africa
* Correspondence: demolamotosho@gmail.com

**Abstract:** The current wave of technological development globally necessitates certain entrepreneurial abilities for most professionals to succeed in the job market. This paper explores the factors influencing entrepreneurial skills acquisition amongst rural university students in Nigeria and South Africa. The strategic goal of setting up rural universities is to enhance the human capital of the host communities. As a result, an attempt is made to pinpoint challenges that stand in the way of achieving this objective. Data was gathered using a quantitative research methodology. A self-designed questionnaire was used to obtain primary data from 1088 randomly selected third-year students. The study's outcome indicates that the acquisition of entrepreneurial competencies amongst undergraduates in the two higher education institutions was impacted by a number of similar factors, such as the university support system, campus entrepreneurial network, family background, availability of mentors, and entrepreneurship education curriculum. The study focuses on the necessity for universities to address skill shortages among undergraduates. Thus, the results could serve as a guide for policymakers on how to motivate rural university students to obtain globally relevant skills.

**Keywords:** entrepreneurship education; entrepreneurial skills; university students; South Africa; Nigeria

## 1. Introduction

Education and skill development operate in synergy like two sides of a coin [1]. This implies that effective higher education is critical to the development of the human capital required for nation-building. In other words, higher education institutions are established to support the development of the increasingly diverse variety of skills needed to fulfil the needs of the digital era. However, a review of the work of Lauder & Mayhew [2] suggests that in some cases, earning a degree from a higher education institution may not necessarily translate to the acquisition of the skills required to support societal development. The foregoing suggests that there is a disconnect between the academic institutions' programmes and the requirements of employers. In an attempt to address this disparity, a group of scholars, such as Wei, Liu, & Sha [3], submit that a functional entrepreneurship education programme has the potential to aid students in developing skills such as creativity, digital communication, innovation, and sustainability-mindedness. This underlines the broad concept of entrepreneurship education, which goes beyond preparing students to launch their own businesses and instead upskills them with the fundamental abilities required to thrive in the dynamic job market.

Globally, the prominence of entrepreneurial education has significantly increased since the advent of the COVID-19 epidemic [4]. But the rising popularity of entrepreneurial education programmes is not devoid of challenges. Such challenges are reflected in the submissions of Ikuemonisan, Abass, Feleke & Ajibefun [5], who opine that the traditional approach of offering entrepreneurship education programmes at many African higher

education institutions raises concerns about their effectiveness. Thus, it is crucial to evaluate the efficacy of the entrepreneurial education programmes offered by higher education institutions, particularly in the sub-Saharan African region. Such evaluation correlates with the notion of Morselli [6] who aver that entrepreneurship education in any institution of learning should be evaluated on a regular basis to determine how well it enables students to develop competencies such as work-ready skills, innovative attitudes, and knowledge applicable to the diverse vocations for the advancement of individuals' social and economic well-being.

The two most populous countries on the African continent are Nigeria and South Africa. If the human capital and mineral resources in Nigeria, South Africa, Angola, Mali, Egypt, Kenya, and Botswana are efficiently utilised, the region could be transformed remarkably [7]. This perspective coincides with the view of Doringer [8] and Scott & Ivala [9] who submit that graduates from higher education institutions should be exposed to experiential learning that will enable them to make significant contributions to nation-building. Surmise to state that the essence of higher education is to develop human capital, which could then translate to societal development. In congruence, Sansone et al. [10] posit that the global economy requires graduates that are equipped with a range of transversal skills and entrepreneurial competencies that will enable them to succeed in practically any productive enterprise.

These entrepreneurial competencies; according to Canton [11], include the capacity to think constructively, solve problems creatively and communicate effectively. The foregoing suggests that graduates require both academic credentials and a range of fundamental skills in order to succeed in the job market. Hence, the overarching challenge herein is to identify the drivers of entrepreneurial skills acquisition amongst university students; this will offer insights for policies aimed at improving levels of entrepreneurship in academic settings, especially in the Sub-Saharan African university contexts. There are currently no comparative studies specifically addressing the issue in the context of developing countries in rural Sub-Saharan Africa. As a result, the study calls for a re-think on how rural universities in the region may be re-positioned to function as agents of change in the digital age. Additionally, the outcome of this study calls for a rethink on how entrepreneurship education could be reimagined and presented in the rural university context.

The remaining sections of this article are organised as follows: Section 2 presents the problem statement; research methodology is covered in Section 3; study findings are presented in Section 4; Section 5 addresses the empirical findings in light of the current situation of rural-based universities in Africa's Sub-Saharan region; and the concluding remarks, policy implications, and directions for further study are presented in within Sections 6–8.

## 2. Rural Universities

Rural universities are strategically positioned to address the needs of their host communities while also stimulating rural community growth [12]. This implies that rural universities are established in predominantly rural communities to help stimulate rural development through innovative research and human capital development. According to Nkomo & Sehoole [13], the approach as a whole provides an opportunity for rural community residents to realise their full potential and contribute significantly to the development of their country. Balfour et al. [14] outline two rationales for the relevance of generative theory of rurality. First, the theory enabled scholars to understand their research findings in a rural context. Secondly, it stimulates rural dwellers to act as agents of change in their local communities. The second reason is indeed relevant for the current study because it conceptualises the rural education institutions as agents of transformation. Hence, rural universities are expected to be adequately funded and managed to carry out their transformative mission effectively.

Conversely, regarding readiness for contemporary social and economic upheavals, especially those brought on by the fourth industrial revolution, rural universities in South Africa seem to be fragile [15]. Similarly, in terms of financing and infrastructure, the majority of Nigerian rural universities demonstrate a significant gap between what is in place and what is required to deliver functional entrepreneurship education [16]. This foregoing suggests that rural universities in South Africa and Nigeria struggle for relevance in terms of fulfilling the purpose of their existence, and the authors contend that since the discourse surrounding the progress of higher education institutions has socio-political connotations, the rurality of these institutions of learning may not be the sole factor contributing to their struggle for relevance.

### 3. Statement of Problem

Although the goals of Entrepreneurship Education attract ongoing debate, frequently its purpose is seen to equip students with the relevant skills, such as creative attitudes, innovative thoughts, and entrepreneurial mindsets, which are necessary for the socioeconomic growth of society as a whole [17]. This implies that higher education institutions are established to support the development of the increasingly diverse variety of skills needed to fulfil the demands of the digital age. However, scholars such as Salem & Mobarak, Ramnund-Mansingh & Reddy and Oanda & Ngcwangu [18–20] posit that despite huge financial allocations and annual expenditure on the higher education sector, the majority of recent graduates from African institutions of learning exhibit inadequate skill levels. This phenomenon, according to Arshi et al. [21] is described as "education for all," a reality that is becoming prevalent in African universities. Consequently, this study's main objective is to determine students' perceptions of factors influencing the acquisition of entrepreneurial skills in the selected university in Nigeria and South Africa.

The following are the study's research questions:

**Research Question 1. (RQ1):** *What are the students' perceptions of factors influencing the acquisition of entrepreneurial skills in the selected university in Nigeria and South Africa?*

**Research Question 2. (RQ2):** *What strategies are employed for teaching entrepreneurship education in the selected Nigerian and South African universities?*

### 4. Methodology

Survey research methodology was adopted for the study. Data was gathered using a quantitative research approach. According to Asenahabi [22], survey research is a way of collecting data using a sample that is representative, which may enhance applicability to a certain group. The two rural-based universities in Nigeria and South Africa were specifically selected because they had comparable characteristics. These characteristics include size, faculties, location, population, and funding source amongst others. A pilot study was conducted so as to test, verify, and refine the research instruments in order to identify problems that the respondents might face in having to understand the questions. During the pilot study, some ambiguous and difficult questions were observed based on participants' views. Hence, the identified items were modified while others were removed.

1480 randomly selected third-year students (740 students from each university) were initially provided with self-report instruments to complete. However, 392 questionnaires were either not returned, or were not fully completed. The remaining 1088 questionnaires resulted in a 73.5% response rate. The 1088 participants in this study represent approximately 10% of the total population of third/final year students at the two selected universities. Due to their age range, exposure, and on-campus experiences, third-year students, who were primarily between 21 and 30 years old, were chosen. 512 respondents from a university in South Africa, and 576 from a university in Nigeria made up this sample population. The participants were selected at random to complete a structured questionnaire on factors that make acquisition of entrepreneurial skills on campus challenging for students.

The five-point Likert scale questionnaire was divided into two sections, the first of which was intended to gather information on the respondents' personal demographics,

and the second section focuses on the study's research objectives. The questions were all created to examine the same variables at the two universities that were selected. This was done to make sure that the comparisons between the two universities were accurate. The questionnaire's items were designed to derive data on the variables influencing the acquisition of entrepreneurial skills amongst the selected university students. Therefore, the questionnaire was created using a scoring system consisting of: Strongly Agree (5 points), Agree (4 points), Unsure (3 points), Disagree (2 points), and Strongly Disagree (1 point). A five-point Likert was chosen over a seven-point scale since it was highly recommended by scholars as it saves respondents' time while increasing the response quality [23]. Table 1 shows how the sample population is represented.

**Table 1.** Respondents' statistics.

| Gender | Nigeria | | South Africa | |
|---|---|---|---|---|
| | Frequency (N = 512) | Percentage | Frequency (N = 576) | Percentage |
| Male | 238 | 46.5% | 276 | 47.9% |
| Female | 274 | 53.5% | 300 | 52.1% |

## 5. Research Results

Using the information gathered from the participants, the findings of the study are presented below:

*RQ1*: What are the students' perceptions of factors influencing the acquisition of entrepreneurial skills in the selected university in Nigeria and South Africa?

The research outcomes are presented as follows.

Table 1 indicates that female respondents outnumbered male respondents in both countries.

According to Table 2, about 1.2% of South African respondents agreed that their university has a support system in place for potential student entrepreneurs, whereas approximately 98 percent of South African respondents did not share this view. In this regard, about 2% of Nigerian respondents believed that their institution provides a support system for potential student entrepreneurs, whereas almost 94 percent disagreed. The majority of respondents from the two institutions generally disclaimed that their institutions provide support systems for potential student entrepreneurs.

**Table 2.** Respondents' perspectives on whether the university has a support system in place for potential student entrepreneurs.

| | SD | D | U | A | SA | Total |
|---|---|---|---|---|---|---|
| | N(%) | N(%) | N(%) | N(%) | N(%) | |
| Nigeria | 482(82.3) | 64(11.5) | 20(3.8) | 8(1.7) | 2(0.7) | 576(100.0) |
| South Africa | 438(86.3) | 55(10.5) | 9(2.0) | 6(0.8) | 4(0.4) | 512(100.0) |
| Total | 920(84.2) | 119(11.0) | 29(2.9) | 14(1.3) | 6(0.6) | 1088(100.0) |

This suggests that the two selected universities' support systems for prospective student entrepreneurs are either non-existent or inadequate. This conclusion supports the work of Ibrahim et al. [24], who asserts that, in most cases, university students in Sub-Saharan Africa do not receive institutional assistance when they embark on entrepreneurial projects. This creates a notion that the students are being trained to operate as job seekers after graduation.

Table 3 shows that 94.9 percent of the South African respondents indicated that they were not part of any entrepreneurial network. On the other hand, 90.3% of Nigerian respondents also signified that they were not part of any entrepreneurial network. It may

be inferred from this that the majority of undergraduates at the two institutions do not participate in any entrepreneurial networks. This conclusion is in line with the findings of Vezi-Magigaba [25], who claim that a major obstacle for student entrepreneurs is the lack of networking and mentorship opportunities, in the context of Sub-Sahara African Universities, there is a lack of information on how to create networks that would be fruitful.

**Table 3.** Respondents' view on whether students belong to any entrepreneurial network.

|  | SD | D | U | A | SA | Total |
|---|---|---|---|---|---|---|
|  | N(%) | N(%) | N(%) | N(%) | N(%) |  |
| Nigeria | 450(78.1) | 70(12.2) | 34(5.9) | 14(2.4) | 8(1.4) | 576(100.0) |
| South Africa | 430(84.0) | 56(10.9) | 16(3.1) | 8(1.6) | 2(0.4) | 512(100.0) |
| Total | 880(80.9) | 126(11.6) | 50(4.6) | 22(2.0) | 10(0.9) | 1088(100.0) |

Table 4 indicates that 3.2% of South African respondents agreed that they had access to entrepreneurship mentors at the university, while 93 percent of the respondents did not believe so. Meanwhile, 4.1 percent of their Nigerian counterparts indicated that they had access to entrepreneurial mentors at the university, whereas the majority of the respondents (89.6 percent) had a contrary view. The foregoing suggests that university students from both countries have limited access to entrepreneurial mentors. This result is consistent with the study of Rapheal [26], who contends that entrepreneurial mentors are rarely available to African students, particularly at universities in sub-Saharan Africa.

**Table 4.** Respondents' views on access to entrepreneurship mentors in the university.

|  | SD | D | U | A | SA | Total |
|---|---|---|---|---|---|---|
|  | N(%) | N(%) | N(%) | N(%) | N(%) |  |
| Nigeria | 456(79.2) | 60(10.4) | 36(6.3) | 14(2.4) | 10(1.7) | 576(100.0) |
| South Africa | 424(82.8) | 50(9.8) | 22(4.3) | 10(2.0) | 3(1.2) | 512(100.0) |
| Total | 880(84.0) | 110(10.3) | 58(3.7) | 24(0.9) | 26(0.1) | 1088(100.0) |

Table 5 shows that about 2% of respondents from South Africa acknowledged having an entrepreneurial family background, whereas 94.2 percent of the respondents did not believe so. On the other hand, 89.3 percent of the Nigerian respondents indicated that they do not have an entrepreneurial family background, and 2.7 percent of the respondents acknowledged having entrepreneurial backgrounds. The results are in line with those of Malebana and Swanepoel [27], who submit that the majority of rural university students in South Africa are from non-entrepreneurial families. Similarly, Ayodele et al. [28] indicate that only a small proportion of Nigerian university students selected for his study had records of family entrepreneurial backgrounds.

**Table 5.** Perceptions of respondents on having an entrepreneurial family background.

|  | SD | D | U | A | SA | Total |
|---|---|---|---|---|---|---|
|  | N(%) | N(%) | N(%) | N(%) | N(%) |  |
| Nigeria | 418(72.6) | 96(16.7) | 46(8.0) | 10(1.7) | 6(1.0) | 576(100.0) |
| South Africa | 412(80.5) | 70(13.7) | 18(3.5) | 10(2.0) | 2(0.4) | 512(100.0) |
| Total | 830(84.0) | 166(10.3) | 64(3.7) | 20(0.9) | 8(0.1) | 1088(100.0) |

Table 6 shows that 1.2 percent of South African respondents believed that university entrepreneurship curricula stimulate innovative thoughts amongst students, however,

97.3 percent of the respondents did not believe so. On the other hand, the majority of Nigerian respondents (61.5 percent) believed that the university entrepreneurship curricula stimulate innovative thoughts amongst students, while 36.5 percent of the respondents had a contrary view. The striking contrast in viewpoints between Nigerian and South African participants in this respect might be linked to the hands-on entrepreneurial experience provided by the selected Nigerian university (see page 7), which appears to be rare at the selected South African university.

**Table 6.** Views of respondents on whether university entrepreneurship curricula stimulate innovative thoughts amongst students.

|  | No | Unsure | Yes | Total |
|---|---|---|---|---|
|  | N(%) | N(%) | N(%) |  |
| Nigeria | 210(36.5) | 12(2.1) | 354(61.5) | 576(100.0) |
| South Africa | 498(97.3) | 8(1.6) | 6(1.2) | 512(100.0) |
| Total | 708(65.1) | 20(1.8) | 360(33.1) | 1088(100.0) |

The finding of the Nigerian aspect of the study is consistent with the work of Ratten & Usmanij [29], who contend that entrepreneurship educators should adopt innovative strategies in their pedagogy, this would stimulate innovative thinking amongst students. Surmise to state that entrepreneurship education is more than simply teaching students how to establish a firm; it is also about improving students' sense of innovation. In other words, entrepreneurship training must be seen through the lens of creativity, innovation, and resourcefulness, amongst other attributes.

Table 7 indicates that about 7 percent of the respondents from South Africa acknowledged that students are offered hands-on entrepreneurship experiences on campus, whereas 89.5 percent of the South African respondents had a contrary view. However, in the Nigerian context, about 62 percent of the respondents concurred that students are offered hands-on entrepreneurship experiences on campus, whereas 33 percent of the Nigerian respondents had a contrary view. This finding corroborates the works of Breed & Mehrtens [30], who aver that students' entrepreneurial skills develop in an exploratory manner when universities embrace a more practical-oriented pedagogy. The foregoing is bolstered by a comparative analysis in Table 8, which suggests that Nigerian respondents are positioned to demonstrate a somewhat greater degree of entrepreneurial aptitude.

**Table 7.** Respondents' views on whether students are offered hands-on entrepreneurship experiences.

|  | No | Unsure | Yes | Total |
|---|---|---|---|---|
|  | N(%) | N(%) | N(%) |  |
| Nigeria | 190(33.0) | 28(4.9) | 358(62.2) | 576(100.0) |
| South Africa | 458(89.5) | 20(3.9) | 34(6.6) | 512(100.0) |
| Total | 648(59.6) | 48(4.4) | 392(36.0) | 1088(100.0) |

**Table 8.** Comparative analysis of the factors in relation to Nigeria and South Africa.

| Criteria | Country | N | Mean Rank | Sum of Ranks | Test Statistics | *p*-Value |
|---|---|---|---|---|---|---|
| University support system for potential student entrepreneurs | Nigeria | 576 | 271.13 | 69,408.00 | 36,512.00 | 0.324 |
|  | South Africa | 512 | 273.72 | 71,832.00 |  |  |
|  | Total | 1088 |  |  |  |  |
| Student affiliation to entrepreneurial networks | Nigeria | 576 | 270.81 | 61,328.00 | 36,432.00 | 0.389 |
|  | South Africa | 512 | 274.00 | 63,912.00 |  |  |
|  | Total | 1088 |  |  |  |  |

**Table 8.** *Cont.*

| Criteria | Country | N | Mean Rank | Sum of Ranks | Test Statistics | *p*-Value |
|---|---|---|---|---|---|---|
| Student access to entrepreneurship mentors in the university | Nigeria | 576 | 274.19 | 76,192.00 | | |
| | South Africa | 512 | 271.00 | 72,048.00 | 36,432.00 | 0.389 |
| | Total | 1088 | | | | |
| Student has entrepreneurial family background | Nigeria | 576 | 274.38 | 70,240.00 | | |
| | South Africa | 512 | 270.83 | 72,001.00 | 36,384.00 | 0.208 |
| | Total | 1088 | | | | |
| Entrepreneurship curricula stimulate innovative thoughts among students | Nigeria | 576 | 278.35 | 65,344.00 | | |
| | South Africa | 512 | 274.42 | 61,323.00 | 31,023.00 | 0.211 |
| | Total | 1088 | | | | |
| Students are offered hands-on entrepreneurship experiences | Nigeria | 576 | 274.44 | 78,256.00 | | |
| | South Africa | 512 | 420.78 | 40,213.00 | 11,681.00 | 0.001 ** |
| | Total | 1088 | | | | |
| *Teaching strategies* | | | | | | |
| Learning-by-doing approach | Nigeria | 576 | 369.20 | 88,915.50 | | |
| | South Africa | 512 | 275.43 | 79,324.50 | 10,019.50 | 0.011 ** |
| | Total | 1088 | | | | |
| Lectures & assignments | Nigeria | 576 | 267.80 | 68,558.00 | | |
| | South Africa | 512 | 246.67 | 65,682.00 | 56,621.00 | 0.004 |
| | Total | 1088 | | | | |
| Case Studies | Nigeria | 576 | 270.88 | 69,345.50 | | |
| | South Africa | 512 | 200.94 | 61,289.50 | 34,419.50 | 0.001 |
| | Total | 1088 | | | | |
| Workshop/Conferences | Nigeria | 576 | 269.96 | 64,110.00 | | |
| | South Africa | 512 | 274.76 | 79,130.00 | 36,214.00 | 0.003 |
| | Total | 1088 | | | | |
| University-Industry Interaction | Nigeria | 576 | 267.58 | 68,501.00 | | |
| | South Africa | 512 | 285.87 | 70,739.00 | 56,205.00 | 0.001 |
| | Total | 1088 | | | | |

The *p*-values with the symbol ** indicates a true statistical difference between respondents' views of the two nations.

*RQ2:* What strategies are employed for teaching entrepreneurship education in the selected Nigerian and South African universities?

The results are presented as follows.

From Table 9 it can be stated that lecture & assignments techniques are the most popular instructional methods of teaching entrepreneurship at the two universities being compared (SA = 97.3%, NG = 70.1%). This suggests that the teaching strategies employed for teaching entrepreneurship education in the two universities are relatively similar. However, the experiential approach to entrepreneurial education is popular in the Nigerian university, whereas this strategy is not employed by the South African university.

The foregoing is affirmed by the majority (59.5 percent) of Nigerian respondents who indicate that they learned entrepreneurship through the learning-by-doing approach. Meanwhile, 98 percent of their South African counterparts indicate that a hands-on approach to entrepreneurship education is not offered in their domain. In addition, other methods of teaching, such as university-industry links and case studies, appear to be

uncommon for teaching entrepreneurship in both of the universities being compared. A critical analysis of all the preceding respondents' views suggests that entrepreneurship education teaching strategies in both universities could be improved. In congruence, Lackeus [31], opines that students are more engaged by experiential learning platforms than lectures. This might explain why some entrepreneurship programmes are more successful than others.

**Table 9.** Views of respondents on the methods employed at the selected universities for teaching entrepreneurship.

| Teaching Methods | Country | No | Unsure | Yes | Total |
|---|---|---|---|---|---|
| Learning-by-doing approach | Nigeria (%) | 201 (34.9%) | 32(5.6) | 343 (59.5%) | 576 (100.0%) |
| | South Africa (%) | 498 (97.3%) | 10(1.9) | 4 (0.8%) | 512 (100.0%) |
| Lectures & assignments | Nigeria (%) | 151 (26.2%) | 21 (3.6%) | 404 (70.1%) | 576 (100.0%) |
| | South Africa (%) | 5 (0.9%) | 9 (1.8%) | 498 (97.3%) | 512 (100.0%) |
| Case Studies | Nigeria (%) | 531 (92.2%) | 25(4.3) | 20 (3.5%) | 576 (100.0%) |
| | South Africa (%) | 449 (87.7%) | 52 (10.2%) | 11 (2.1%) | 512 (100.0%) |
| Workshop/Conferences | Nigeria (%) | 422 (73.3%) | 05 (0.9) | 149 (25.9%) | 576 (100.0%) |
| | South Africa (%) | 220 (42.9%) | 15 (2.9) | 273 (53.3%) | 512 (100.0%) |
| University-Industry Interaction | Nigeria (%) | 536 (93.1%) | 21(3.6) | 19 (3.3%) | 576 (100.0%) |
| | South Africa (%) | 486 (94.9%) | 12(2.3) | 14 (2.7%) | 512 (100.0%) |

The Mann-Whitney U-statistic was used to compare the opinions of South African and Nigerian respondents on each of the criteria on a pair-wise basis. This statistic is the standard two-sample t-nonparametric test's counterpart. From Table 8, it can be deduced that responses from the two nations have some noticeable similarities. The *p*-values of the statements, which are all less than the significance level of 0.05, support this assertion, with the exception of responses that pertain to the learning-by-doing approach to entrepreneurship education where the *p*-value 0.11 is higher than 0.05. In this regard, the results from the two nations differ statistically. The comparative analysis indicates a true statistical difference between respondents' views of the two nations. Surmise to state that the experiential learning approach is incorporated into the entrepreneurship education curriculum at the university in Nigeria, but this is not the case in the South African university.

## 6. Discussion of Findings

The results of this study suggest that the variables impacting students' acquisition of entrepreneurial skills at Nigerian and South African rural-based universities are similar to a large extent. The identified flaws in key components of the two selected universities' entrepreneurship training systems may impede the overall production of new entrepreneurs. A fundamental flaw in this regard is the lack of support for potential student entrepreneurs. The findings indicate inconsistency and lack of significance in the coefficients relating to the factors that support entrepreneurial skills acquisition in the two university settings.

In this regard, universities in Sub-Sahara Africa could do more to support the growth of entrepreneurship on their campuses. Failure to set incentives and succour for entrepreneurial-minded students may render the quality of learning in the university system redundant. Universities and other higher education institutions need to develop job creators rather than job seekers in the present economic context. The outcome of the study further indicates that the two selected universities still have a long way to go in encouraging greater levels of student entrepreneurship. The authors contend that support structures that stimulate entrepreneurial skills acquisition in a university setting seem to be lacking in this regard, and this condition poses significant hurdles for entrepreneurship policy.

Furthermore, family background is a crucial limiting factor in this study. In congruence, Schimperna, Nappo & Marsigalia [30] posit that entrepreneurial family background stimulates students to embark on entrepreneurial projects and vice versa. According to Lanphier & Carini [31], entrepreneurial family members may provide a prospective entrepreneur with relevant information, monetary resources, cash management skills, and assurances. In other words, financial support from a relative could enable a prospective student entrepreneur to experiment with entrepreneurial projects, however, the forgoing is not without criticism; Geza et al. [32], discover a relationship between entrepreneurial family background and lower rates of self-employment.

One of the key challenges highlighted by these research findings is students' limited access to mentors. According to Doringer, mentorship advantages include increased managerial abilities, expanded vision for commercial endeavours, and the capacity to spot new prospects [33]. Furthermore, mentorship programs educate students on what it means to be an entrepreneur, and in addition introduce students to business networks [34]. Due to effective collaboration and networking between students and other members of the business environment, students are better able to recognise business possibilities, which increases their chances of succeeding in the development of new firms [35–37]. In the context of Sub-Saharan African universities, programmes that include students in external networks might be created to encourage prospective student entrepreneurs.

With regard to teaching strategies, the results indicate that an experiential approach to entrepreneurship education has proven to be more successful than the traditional lecture method. Such hands-on activities include; outreach adventures, and business competitions in addition to the basic offering of entrepreneurship courses [38]. The authors contend that higher education institutions may inspire student entrepreneurs by promoting a network of innovators that foster enterprising behaviours. This scenario entails learning by doing, learning from failures, and learning from entrepreneurs.

Despite the identified inflexibilities in this regard, entrepreneurial training at the two selected universities still remains a prospect for progress. A thorough understanding of the system's weak links might aid policymakers in charting the way forward. Above all, this study calls for a critical review of entrepreneurial training models in the two selected universities. It is critical for higher education institutions to equip students with core skills in addition to academic knowledge, such relevant skills may enable students to operate as successful entrepreneurs or intrapreneurs upon graduation.

## 7. Concluding Remarks

This study has explored the issue of factors influencing entrepreneurial skills acquisition amongst students in the context of rural universities in Nigeria and South Africa. The outcome offered vivid insights into the dynamics of the university support system, family background, teaching strategies, availability of mentorship, and networking opportunities in the two selected universities, this insight has been derived from a strategically sampled group of students in these countries. Despite the necessity for universities to play a crucial role in sustainable development by developing citizens' entrepreneurial capacities, there seems to be a long way to go in ensuring that the selected rural universities, in this regard, take their position as agents of change.

The authors contend that, aside from South Africa and Nigeria, the results of this study might reflect the current reality of rural universities in other African nations. Nonetheless, the likelihood of exceptional instances of successful entrepreneurship training models on the African continent is acknowledged. As a result, a quick fix may not be sufficient to resolve the complex inefficiencies revealed in this regard, rather, it necessitates a comprehensive rethinking of how the academic environment functions. This comparative research might pave way for the development of models that may influence future policymaking endeavours in the Sub-Saharan African higher education sector. Some recommendations are offered in this regard. If properly examined and implemented, they are designed to provide the optimal environment for functional entrepreneurial education to thrive.

Finally, this study does not go without limitations. The data provided by participants came only from one stakeholder group, i.e., students. For a more complete picture of the challenges of delivering entrepreneurship education in rural locations in Africa, other stakeholders such as educators, higher education leaders, and community representatives could be included. Furthermore, due to fear of the unknown, not all university students have the courage to disclose the shortcomings within their institution of learning to the general public. This may have also limited the accuracy of the information provided. Moreover, the structured questionnaire used in this study may have narrowed the participants' viewpoints.

## 8. Recommendations

Sequel to the outcomes of this study, the following recommendations are made as follows:

- Traditional teaching approaches could be modified so that entrepreneurship education is adapted to students' sociocultural demands. It is also imperative to integrate entrepreneurship education into all fields of study at the university level.
- The concept of connecting university students to relevant role models could be considered; this strategy has the capacity to revive students' entrepreneurial spirit.
- Universities could create a small business development section to aid university students. This will contribute to the development of an entrepreneurial hub within a university and its surrounding communities.
- Universities could ensure that entrepreneurship education programmes draw on contemporary societal challenges in their domains.
- Educators and practitioners involved in entrepreneurship development programmes could provide students with a variety of learning opportunities.
- Rural universities could upgrade the quality of their entrepreneurial training through the use of live projects for teaching and learning purposes.

## 9. Suggestion for Further Study

The choice of two higher education institutions in Nigeria and South Africa may not offer a thorough understanding of the influencing factors on entrepreneurial skills amongst university students across the African continent. As a result, it is advised that additional in-depth studies on this subject be conducted using several rural institutions in various geographical regions of Africa. This will either confirm or disprove the findings of this study.

**Author Contributions:** Methodology, M.A.M.; Resources, K.M.K.; Data curation, K.M.K.; Writing—original draft, O.A.O.; Writing—review & editing, M.A.M. All authors have read and agreed to the published version of the manuscript.

**Funding:** This research received no external funding.

**Institutional Review Board Statement:** The study was conducted in accordance with the Declaration of Helsinki, and approved by the University of Zululand Research Ethics Committee (UZREC) (protocol code:18/525; date of approval: 19 September 2019).

**Informed Consent Statement:** All procedures performed in studies involving human participants were in accordance with the ethical standards of the University of Zululand. Informed consent was obtained from all individual participants included in the study.

**Data Availability Statement:** Not applicable.

**Conflicts of Interest:** The authors declare no conflict of interest.

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
