# Peer review of "Comparative Factors Influencing Entrepreneurial Skills Acquisition amongst Students in Rural Universities of Sub-Sahara Africa’s Developing Nations"

_education, doi:10.3390/educsci13030229_

Round 1

Reviewer 1 Report

I am pleased to have the opportunity to review this research paper. This study attempted to explore Comparative Factors Influencing Entrepreneurial Skills Acquisition amongst Students in Rural Universities of Sub-Sahara 3 Africa's Developing Nations.

The topic of this research study is interesting and fits within the journal scope, I think authors should apply the comments indicated to increase the quality of research justification, contributions and findings. What is the originality of this research?  Paper research gap and originality should be better presented at the end of introduction section

the topic of this research study is interesting and fits within the journal scope, I think authors should apply the comments indicated to increase the quality of research justification, contributions and findings.

The authors follow a scientific structure appropriate to the work. There is research question. They answer the research questions. Appropriate methodology for the study. Authors referenced in the literature review justify the results.

Very good article. Would like to see only the originality of the article.

Reviewer 2 Report

Thank you for submitting your paper. Overall it addresses an important topic though I feel your conclusions extend beyond what is warranted based on the data available to you. I hope you find the below useful in refining your paper before publication.

The general build up of the paper is fine: the expansion of entrepreneurship education (EE), how it supports individuals and by extension society etc. are all well-trodden paths. There is some novelty in the focus on an African, and here specifically Nigerian and South African context, although increasingly we are seeing studies based in Africa (which is a good thing).

The paper is largely written well, though at times I felt some statements lacked depth or were quite sweeping, e.g. “Entrepreneurship education is globally acknowledged as a process of stimulating students to develop desirable traits“ – desired by whom and what form of education tries not to develop ‘desirable traits’? You include as an example the desirable trait of a sustainability-mindset – I don’t think this is common across all entrepreneurship programmes.

Although quite broad, I really like your research questions. Because of their breadth I’m not sure how a quantitative approach would be able to meaningfully address them.

How does the focus on obstacles “The participants were chosen at random to provide open-ended responses to questions on the obstacles they face in their bid to develop entrepreneurial skills on campus” relate to either RQ1 or RQ2?

I also wonder why you focus on students (only) to answer the research question (RQ2) about “What strategies are employed for teaching entrepreneurship”? Will the students be experts on teaching strategies or the teachers?

You should also clarify non-response rates and any potential bias you feel might characterise your sample.

Given that you make a great deal of the fact that these are rural universities in the abstract I was surprised the notion of rurality (or at least what characterises rural universities and how this relates to the delivery of EE) was hardly discussed subsequently.

In Table 2, what do you mean by “a support system in place for potential student entrepreneurs” and would respondents have understood what you meant? The same goes for the content of Table 3: what does ‘belonging to any entrepreneurial network’ mean? Would students share an understanding of what this means? What are the implications for the validity of your measures?

You suggest in the methodology that you use a 4-point scoring system but the tables include five points (I’m guessing ‘U’ stands for ‘unsure’ this being your fifth item on your scale).

The results of Table 6 are quite astonishing given the dramatic difference between Nigerian and South African respondents – could you explain the difference?  

You also need to be very careful with your wording as you interpret the tables. For example, Table 2 asks whether the university has a support system in place but you then go on to interpret this as students responding to whether the support system is sufficient – this is not the same thing as having one.

The discussion is helpful and more accurate generally in terms of wording than the findings. I still find elements of the discussion moving beyond (far beyond) what is warranted based on the data though. For example, the claim that “the quality of entrepreneurial training in the two selected universities falls below global expectations” is far too sweeping. Firstly, I’m not convinced the study accurately measures the quality of entrepreneurial training (how have you operationalised ‘quality of entrepreneurial training’) and secondly, is there a global expectation when it comes to its quality?.

You claim “Hence, the primary function of educational programs is to raise student knowledge and promote the entrepreneurial path as a viable career option“ and yet earlier in the paper you recognise that EE goes beyond preparation solely for business start up. In some countries EE does focus heavily on business start-up, but in many countries EE is primarily about developing enterprising skills, whether these are used to set up a business or to employ them within an organisation (intrapreneurship).

There is no real discussion of the study’s limitations.

The conclusion needs to be completely written to reflect the contents of the paper. The claims the paper currently makes in the conclusion go far beyond what the data suggest. I’m sorry, but to claim “This study has uncovered the determinants of entrepreneurial skills acquisition amongst university students in the context of rural Sub-Sahara Africa,” is not warranted based on your survey and its size. What your study does is provide some indication as to the nature of EE provision in sub-Saharan Africa and this is welcome.

Reviewer 3 Report

The article is interesting and brings out important issues! Congratulations!

I have some suggestions to further improve it.

Regarding the RQ1, I think it can be reformulated. The paper does not answer what factors influence the acquisition of entrepreneurial skills. The paper describes students’ perceptions to factors that the authors previously identified as important to the development of entrepreneurial skills. There’s a big difference. That does not mean that the paper failed to achieve its goal. It means that the research question needs to be rewritten. 

Still, the last sentence of the “statement of problem” session says that ˜this study’s main objective is to determine the variables that restrict university students from gaining 21st century skills”. I think that should also be reformulated. You are not determining variables. You are describing the entrepreneurial context. That should be the paper’s main goal. Also, when you state that they “restrict” students it seems like a conclusion before time. 

Regarding the methodology session, I would recommend that you make some things clearer. For example: which items were removed or modified and why? Give at least one example. How were the questions elaborated? Were they based in other studies? It is important that the readers know that. Why were 392 questionnaires discarded? Why did you choose a 5-point Likert scale instead of a 7-point or 10-point one? It is important to explain that, considering the literature. 

Self-designed is not the best word to describe the questionnaires. I prefer self-report instrument.

Regarding the results session:

I think the tables are not very clear (you should write the country’s name instead of only the abbreviation) and I think the tables take a lot of space. Maybe only table 9 is enough.

Some of your conclusions are inadequate. For example: “it does suggest that the female population is likely to be greater than the total male population”. That is not true. That is a fact only for your sample.

In the paragraph about table 4 you state that “This result is consistent with the studies of Villa et al., (2022), who contend that resource and time limitations often restrict students' access to mentors.” That is imprecise, because your instrument did not measure students’ time and resources available, so you cannot jump into that conclusion. 

The same is true for family background. As you did not measure entrepreneurial intention or make any causal relationships between those measures, you cannot affirm that the lack of entrepreneurs in the family make them less entrepreneurial, as you don’t have causal proof for this sample.

Still, some statements sound like personal opinion, like “entrepreneurship education methods in both universities seem to be ineffective, in other words, the methods lack a creative edge.” You can try to find another way of saying that. 

In the concluding remarks, I suggest you delete the word small to refer to your sample size. I think it is a good sample size.

As to the recommendation’s session, you can try to make them sound less imperative and seem more as suggestions. You can do that by changing the verb “need” for “could”, for example. 

I also advise you to split some paragraphs alongside the text, as some of them seem too long. 

Nonetheless, I congratulate you for the work and hope the suggestions help the paper be even better! 

Author Response

Comments uploaded

Round 2

Reviewer 2 Report

Thank you for addressing my concerns. I am happy with the revisions and note only a few comments/minor changes as follows:

Page 2: “Balfour et al 2008 outline two rationale” should be ‘rationales’

 Page 3: While it is good to see the inclusion of the reference to Subrahmanyam (2020) I would still argue that the goals of entrepreneurship education are complex and debated. Therefore, I would precis the sentence “The goal of entrepreneurship education is to equip students…” with the following, “Although the goals of Entrepreneurship Education attract ongoing debate, frequently its purpose is seen to equip students…”

It is good to see the authors try to interpret their results now, especially where there is a big contrast between the two locations.

 Conclusion: “Additionally, the data provided by participants were based on students' perceptions, which may have led to information bias.“ This point is valid though I think could do with a little further explanation, e.g. consider rephrasing to “Additionally, the data provided by participants came only from one stakeholder group, i.e. students. For a more complete picture of the challenges of delivering entrepreneurship education in rural locations in Africa, other stakeholders such as educators, higher education leaders and community representatives could be included”.

Please make sure all references are fully completed, e.g. the Balfour reference does not include page numbers.

Author Response

Comments uploaded
